# Population Movement and Poliovirus Spread across Pakistan and Afghanistan in 2023

**DOI:** 10.3390/vaccines12091006

**Published:** 2024-09-01

**Authors:** Irshad Ali Sodhar, Jaishri Mehraj, Anum S. Hussaini, Muhammad Aamir, Jahanuddin Mahsaud, Shabbir Ahmed, Ahmed Ali Shaikh, Asif Ali Zardari, Shumaila Rasool, Shoukat Ali Chandio, Erin M. Stuckey

**Affiliations:** 1Emergency Operation Center Sindh, Government of Sindh, Karachi 75510, Pakistan; ahmed.ali@rizconsulting.biz (A.A.S.); sindh.nstopteamlead@gmail.com (S.R.); 2Integral Global Health Inc., Islamabad 44000, Pakistan; jaishrimehraj@gmail.com; 3Harvard T.H. Chan School of Public Health, Boston, MA 02115, USA; anumhussaini@hsph.harvard.edu; 4World Health Organization (WHO), Islamabad 45500, Pakistan; amuhammad@who.int (M.A.); sahmed@who.int (S.A.); asifa@who.int (A.A.Z.); 5United Nations International Children’s Emergency Fund (UNICEF), Islamabad 44050, Pakistan; jahanuddin@unicef.org (J.M.); shoali@unicef.org (S.A.C.); 6Riz Consulting, Islamabad 44000, Pakistan; 7National Stop Transmission of Polio (N-STOP) Program, Karachi 75510, Pakistan; 8Bill & Melinda Gates Foundation, Seattle, WA 98109, USA; erin.stuckey@gmail.com

**Keywords:** Afghanistan, Pakistan, polio, immunization, high-risk populations

## Abstract

Population movement dynamics are a critical part of understanding communicable disease transmission patterns and determining where, when, and with whom to deliver appropriate prevention interventions. This study aimed to identify the origin of the Afghan population and their patterns of movement within Karachi, to assess the polio vaccination status of children under the age of five, and to investigate the travel history and guest arrival patterns of individuals from Afghanistan and other regions known to be affected by wild poliovirus type 1 (WPV1) within the past six months. A cross-sectional survey was conducted in selected 12 union councils of Karachi, Pakistan. The data were collected through interviews with Afghan household members and from the frontline workers (FLWs) responsible for the polio vaccination of the children of the same households. Cohen’s kappa was used to check the agreement between information provided by the household participant and FLWs. A total of 409 Afghan household members were interviewed. Travel of any household member outside the city within the last six months was reported by 105 (25.7%) participants, 140 (34.2%) hosted guests within the last six months, and 92 (22.5%) participants reported that guest children were vaccinated in their households. A total of 230 (56.2%) participants observed polio teams at relatives’ households within Karachi, and 127 (31.1%) observed polio teams at relatives’ households outside Karachi in different districts of Pakistan and Afghanistan. Fair to moderate agreement was observed between information provided by the household members and FLWs on the variable’s duration of living at current residence (Kappa = 0.370), travel history (Kappa = 0.429), guest arrival (Kappa = 0.395), and household children vaccinated for OPV (Kappa = 0.419). Substantial population mobility was observed between Afghanistan and Pakistan as well as significant movement of the Afghan population within Karachi in the last six months. These findings warrant attention and targeted implementation of interventions to enhance and sustain both routine and supplementary immunization activities within this demographic group.

## 1. Introduction

Wild poliovirus (WPV) infection leads to paralysis (poliomyelitis) in the at-risk population of children under 5 years of age, significantly impacting the lives of the affected children and their families [1]. Children affected by polio are most often from low- to lower-middle-income families living in areas with poor water and sanitation systems and weak health infrastructure and are therefore often un- or under-immunized with routine immunization antigens including polio. According to the World Health Organization (WHO), polio cases have decreased by over 99% since the launch of the Global Polio Eradication Initiative (GPEI) in 1988, from an estimated 350,000 cases in more than 125 endemic countries to just 12 cases in 2023 in the two remaining endemic countries of Afghanistan and Pakistan. To eliminate polio from its territory, the government of Pakistan declared polio a national emergency in 1994 and currently implements the program managed by Emergency Operation Centers (EOCs) at the national level and in each of its provinces [2]. The government of Pakistan spearheads the EOC network. It acts as a coordination function for GPEI partners WHO, Rotary International, the US Centers for Disease Control and Prevention (CDC), UNICEF, the Bill and Melinda Gates Foundation, and Gavi, the Vaccine Alliance (GAVI) [3].

Main polio program elements in Pakistan include routine immunization (RI), nationwide vaccination campaigns with bivalent oral polio vaccine (OPV) and inactivated polio vaccine (IPV), surveillance to detect signs of acute flaccid paralysis (AFP) in children and detect any poliovirus in the environment, and communications and community engagement to ensure a high level of vaccine acceptance. Routine immunization (RI) against six diseases including poliovirus was commenced in 1978 with the introduction of the National Expanded Programme on Immunizations (EPI) in Pakistan [4]. RI services are given at fixed locations or immunization centers and with minimal outreach by the vaccinators. Besides four doses of the routine OPV within RI services [5], Supplementary Immunization Activities (SIAs) including national and sub-national immunization activities (NIDs and SNIDs) are the main strategy to enhance OPV coverage. The Pakistan National Polio program organizes these SIAs, which last from 5–7 days and in which all households with children under five years of age are visited in 5 days and households where children were not available or refused during the 5 polio campaign days are revisited in the remaining 2 days [6]. During NIDs, mass immunization campaigns are conducted across the country whereas, during SNIDs, campaigns are organized in selected high-priority areas or districts [7]. Also, to ensure vaccination of all travelers against poliovirus, the Pakistan Polio Eradication Programme has set up a Permanent Transit Point for Polio (PTP) at the Pak–Afghan border [8].

Since only one in 200 infections leads to paralysis, the Pakistan polio program takes samples using various methods described elsewhere from sewage sites to isolate any virus that may be circulating in the population [9]. This is to give the program time to conduct targeted vaccination campaigns with high enough coverage levels before transmission levels become high enough to result in paralysis.

In September 2023, wild poliovirus type 1 (WPV1) was isolated from four environmental surveillance (ES) samples collected from different sites in Karachi, Pakistan. All four isolates of WPV1 were linked to the same ES sample collected in Jalalabad, Nangarhar, Afghanistan collected in November 2022: ES-AFG-Nangarhar-Jalalabad-Nov2022. Among these four samples, one was collected from the Haji Mureed Goth ES site in Liaqatabad town, district Central Karachi on 7th September 2023 whose WPV1 isolate was a 98.78% match to the Jalalabad sample. The second sample was collected from the Hirjat Colony PIDC Nalla ES site in Saddar town, district South Karachi on 13th September 2023 whose WPV1 isolate was a 98.56% match with the Jalalabad sample. The third sample was collected from the Machhar Colony ES site in Gadap town, district East Karachi on 12th September 2023; whose WPV1 isolate was a 98.78% match with the Jalalabad sample. The fourth sample was collected from the Sohrab Goth ES site in Gadap town, district East Karachi on 12th September 2023 whose WPV1 isolate was a 98.89% match with the Jalalabad sample [10]. The investigations into the genetic relatedness of poliovirus isolates are conducted in a regional reference laboratory for poliomyelitis at the National Institute for Health (NIH), Islamabad, Pakistan [11]. The regional reference laboratory for polio follows WHO-recommended procedures for detecting and characterizing polioviruses from the stool samples of AFP cases and sewage water samples collected from the environment [12]. The reports of WPV1 detection from any sample are shared internally within the polio program network of Pakistan [10].

Following the notification of these four positive samples, a team from the Pakistan Polio National Emergency Operations Center (NEOC) joined the Sindh EOC in Karachi to determine the most likely reason why the September samples in districts Central, South, and East were all linked to the same virus. An initial investigation was conducted in the drainage Union Councils (UCs) and adjoining UCs of ES sites, which were positive for WPV1 isolates. The team investigated the hypothesis that the ES sites share the same drainage UCs (i.e., sewage from the same/overlapping population is draining into different sites). Still, no evidence of overlapping of ES UCs/sewage channels was noted. They also investigated the possibility of composite sampling where either the individuals collecting the samples used the same sample to represent different sites or used the same collection materials to collect various samples. Still, no evidence of composite sampling was found. The final hypothesis investigated by the team was that the population from Nangarhar infected with WPV1 settled in UCs in different districts that drain to different ES sites. This arrived as the most likely scenario based on the continued influx of Afghans to Karachi and the events leading up to the September sample collections due to gatherings within the Afghan community and the disbursement of Afghans from camps that were not previously part of a drainage system as part of a government of Pakistan crackdown order on undocumented migrants from Afghanistan. However, the team could not definitively rule out that the virus could have been circulating among the community within Karachi [10].

Population movement dynamics are critical to understanding communicable disease transmission patterns and determining where, when, and with whom to deliver appropriate prevention interventions. The polio vaccination status of children under the age of five within the Afghan population residing in Karachi is found to be lower than the population’s average coverage of 95% [13]. This raises concerns as it positions this population as potential carriers of WPV1 actively shedding the virus into the environment. Information such as the vaccination status of children under five years of age belonging to the Afghan population residing in Karachi and population movement patterns is important to determining whether the source of infection is the population recently arriving in Karachi from Afghanistan or whether there is another population in Karachi with low immunity levels who had recently moved from a different area in the same city. Without this key information, the polio program will be more likely to miss opportunities to vaccinate children who reside in or travel to areas with ongoing transmission and risk further spread of the virus when they travel to other locations. To plan a strategy to effectively reach those most at risk for polio with vaccines, the Sindh EOC determined they needed additional information about the target population and to test the hypothesis that WPV1 arrived in Karachi when a specific population from WPV1-infected districts in Afghanistan arrived around the same time, settling within the established Afghan population within the city all across different districts of Karachi that drain into different ES sites.

There were three study objectives. The first was to identify the origin of the Afghan population and their patterns of movement within Karachi and employ social profiling to determine the social characteristics of the Afghan population in Karachi, including linguistic and tribal affiliations. The second was to assess the polio vaccination status of children under the age of five within the study population through comprehensive epidemiological investigations. Finally, the study aimed to investigate the travel history and guest arrival patterns of individuals from Afghanistan and other regions known to be affected by WPV1 within the past six months to ascertain potential links between travel from WPV1-infected areas and the occurrence of poliovirus within the studied population in Karachi.

## 2. Materials and Methods

### 2.1. Study Design, Settings, and Population

A cross-sectional survey was conducted in selected areas of the Karachi division, Sindh, Pakistan. Karachi is a cosmopolitan city with a wide variety of populations across the city. Serving as both the capital city of the Sindh province and the largest urban center in Pakistan, Karachi is segmented into seven districts with 201 union councils. As per the 2017 census, the overall population of Karachi is recorded at 14.91 million, including 2.6 million children under the age of five. However, unofficial estimates indicate the population could be much higher.

The survey was conducted within UCs with a high proportion of the Afghan population residing and encompassing areas that drained into environmental sample collection sites. The study population was a cohort of 61,231 Afghan children under the age of five, as documented in various districts of Karachi (polio program resources Aug 2023). To include enough of the targeted population to meet the sample size (detailed below), twelve UCs were chosen for inclusion in this survey including Gujro UC 4 zones A, B, and C, and Safora UC 13 in District East, Haji Mureed Goth UC 44 and Pahar Gunj UC 20 in District Central, Sultanabad UC 2 in Keamari district, Baloch Goth UC 13, Mominabad UC 1, Islamia Colony UC 9, Qasba Colony UC 8, and Songal UC 5 in District West. These UCs are rural areas with wide population movement and suboptimal basic infrastructure services, especially in the health sector. The population residing in these areas therefore has limited options for their health issues. Due to very few public health institutions, the health-seeking behavior mostly relies on private practitioners and informal healthcare providers [14].

### 2.2. Sample Size Calculation

We calculated the sample size for both the registered Afghan children population and the estimated total Afghan population in Karachi through the Open EPI online software [15]. A total of 61,231 Afghan children under five years old were registered in Karachi. It is estimated that 17% of the total population is under five years old. Therefore, the estimated 360,183 (61,231 × 5.89) Afghan population resides in Karachi. With a population size of 61,231 children under five years of age, 95% confidence interval, and 50% frequency of outcome factor, the minimum sample size required is 382 participants. With a population size of 360,183 participants, 95% confidence interval, and 50% frequency of outcome factor, the minimum sample size required is 384. After a 10% increase in anticipated non-response (38.4 participants), the final sample size required for the survey was 423. The division of sample size per UC was based on the proportion of Afghan households at the geographic locations.

### 2.3. Data Collection

The information was collected through interviews with Afghan household members and from the frontline workers (FLWs) responsible for the polio vaccination of the children of the same households. A comprehensive questionnaire was developed in English; to ensure linguistic clarity, the questionnaire was translated into Urdu to optimize comprehension for interviewers. The questionnaire underwent a pre-testing phase within a similar population to ascertain its relevance and efficiency and identify and address any issues related to language nuances, interviewer understanding, and time constraints during interviews. The questionnaire consisted of two distinct sections: Part I with information garnered through interviews with Afghan household members, and Part II with information from FLWs. UC-level Pashto-speaking workers in the polio program who are familiar with the area and the population were selected to conduct the interviews. Before the initiation of data collection, candidates underwent comprehensive training on the questionnaire. Each data collector was assigned 30 households for interviews. In this study, one questionnaire/response was completed per household. One participant represents one household. The data were collected over five days following the conclusion of the outbreak response campaign (OBR) held from 30 October to 5 November 2023, in Karachi. In this study sample, 30 (7.3%) households were covered by a special mobile team (SMT) strategy, and 379 (92.7%) were covered by a community-based vaccination (CBV) strategy. Campaign duration is the same for both CBV and SMT UCs, but FLWs are hired on a monthly basis in CBV strategy and in SMT areas they are paid for 7 campaign days, as described elsewhere [16,17].

### 2.4. Study Variables

Information was collected on different socio-demographic variables including age and gender of the participant, language, tribe, guest arrival in the last six months, and recent travel to Afghanistan and other districts of Pakistan with a history of WPV1 transmission including Killa Abdullah, Peshawar, and other districts of Khyber Pakhtunkhwa province. In addition, information was also collected on the recent visit of the polio team including the behavior of polio team members with the community and the vaccination status of children under five years old. Information about recent travel and the arrival of guests within the last six months, language, tribe, origin district, and number of children was also collected through the interviews of FLWs. As per the RI schedule in Pakistan, immediately after birth, children receive Bacille Calmette–Guérin (BCG), the first dose of the oral polio vaccine (OPV0), and HepB. At 6 weeks, children receive OPV1, Rotavirus1, the first dose of the pneumococcal conjugate vaccine (PCV1), and pentavalent. At 10 weeks OPV2, Rotavirus2, PCV2, and Pentavalent2. At 14 weeks, children receive OPV3, the first dose of the inactivated polio vaccine (IPV1), PCV3, and Pentavelant3. At 9 months of age, the child receives the first dose of the measles-containing vaccine (MCV1), IPV2, and typhoid. MCV2 is given at 15 months age [5]. Routine immunization coverage is classified into fully vaccinated children who received all the recommended doses in EPI. A partially vaccinated child is a child who started to receive the vaccine through a routine immunization program but could not complete the course with recommended vaccines under EPI. An unvaccinated child is a child who did not receive a single dose of vaccines through a routine immunization program.

### 2.5. Data Management and Analysis

Questionnaires were filled in on hard-copy forms and were subsequently transformed into an electronic format through entry into Microsoft Excel. Rigorous validation checks and thorough data cleaning procedures were carried out to ensure data accuracy and integrity. The refined dataset from the Excel sheets was exported and analyzed using IBM SPSS Statistics version 22 software. Cohen’s kappa is used to see the agreement between information provided by the household participant and FLWs. Kappa is a statistical coefficient representing a statistical classification’s degree of accuracy and reliability. It measures the agreement between two raters who each classify items into mutually exclusive categories. The interpretation of Kappa statistics is 0.01–0.20 slight agreement, 0.21–0.40 fair agreement, 0.41–0.60 moderate agreement, 0.61–0.80 substantial agreement, and 0.81–1.00 almost perfect or perfect agreement.

### 2.6. Ethical Considerations

Ethical approval was obtained from the Provincial Bioethics Committee (PBC) of the Director General Health Services (DGHS), Government of Sindh. All study procedures were explained to all participants in Urdu and Pashto languages, and written informed consent was obtained from each participant. To maintain anonymity, the questionnaires administered for data collection were devoid of any personal identifiers, such as names or NIC details. Data obtained from the survey were exclusively utilized for research purposes, with strict prohibitions on reproduction. Rigorous confidentiality measures were implemented, employing identification numbers or unique codes for each study participant. Hard copies of all research documents were saved under a lock and key, and soft copies were saved in a password-protected hard drive with access restricted to the research team.

## 3. Results

A total of 409 participants (Afghan household family members) were interviewed. A total of 181 participants (44.3%) were from District West, followed by 147 (35.9%) from District East, 51 (12.5%) from District Central, and 30 (7.3%) from District Keamari. The mean age of the participants was 36.43 with an 11.42 standard deviation (SD), which includes a minimum of 18 years to a maximum 80-year-old participant. The majority of respondents (354 (86.6%)) were females. Languages spoken by participants were 298 (72.9%) Pashto, 84 (20.5%) Dari, 20 (4.9%) Turkman, and 7 (1.7%) Uzbek. A total of 63 participants (15.4%) belonged to the Suleman Khel tribe, followed by 48 (11.7%) Aka Khel, 33 (8.1%) Uzbek, and 33 (8.1%) Tajik. A minimum of zero and a maximum of nine children under five years of age were reported in the households. Respondents reported 395 (96.6%) children received OPV during the last campaign. Respondents reported polio teams comprising of one member visited 145 (35.5%) households, teams with two members visited 215 (52.6%), teams with three members visited 46 (11.2%) households, and a four-member team visited 3 (0.7%) households.

Most Afghan household members (331, 80.9%) had lived for ten or more than ten years in their current location residence, 12 (2.9%) for five to ten years, 20 (4.9%) for three to five years, 33 (8.1%) for one to two years, and 13 (3.1%) for less than one year. Additional demographic characteristics of the study are shown in Table 1. When asked where in Afghanistan they lived before coming to Pakistan, 55 (13.4%) mentioned only the country of Afghanistan but did not specify the district of origin. The most frequently mentioned districts were Kunduz (39, 9.5%), Kabul (20, 4.9%), Kandahar (18, 4.4%), Mazar Sharif (14, 3.4%), Helmand (12, 2.9), and Takhar (10, 2.4%). The full list of districts is available in Figure 1a. One participant (0.2%) indicated that they were living in Iran before arriving in Karachi. The most frequently reported origin from other districts within Pakistan included among the majority of 71 (17.4%) were reported from Quetta, followed by 22 (5.4%) from Peshawar, 9 (2.2%) from Zhob, 6 (1.5%) from Pishin, 3 (0.7%) from Waziristan, 2 (0.5%) from Bannu, 2 (0.5%) from Chaman, and 2 (0.5%) from Hyderabad (Figure 1b). A total of 16 (3.9%) participants did not provide information about the district of origin, whereas 56 (13.6%) reported that they were from Karachi (Appendix A).

Among study participants, 234 (57.2%) reported that their children completed their full vaccination course of routine immunization, 158 (38.6%) reported their children received some but not all vaccines, and 17 (4.2%) were not vaccinated by any routine immunization antigen.

The majority of FLWs 207 (50.6%) were working within the area for polio program for 3 to 5 years, followed by 97 (23.7%) for one to two years, 92 (22.5%) for more than five years, 10 (2.4%) for less than 6 months, and 3 (0.7%) for six months to one year. The behavior of the FLWs/polio team with the family was reported as good by 398 (97.3%) participants, satisfactory by 3 (0.7%) participants, and 8 participants (2%) reported bad behavior of the polio team.

Travel of any household member outside the city within the last six months was reported by 105 (25.7%) participants, 140 (34.2%) participants reported they hosted guests who arrived within the last six months, and 92 (22.5%) participants reported that guest children were vaccinated in their households. For the questions assessing travel of household members, 230 (56.2%) participants observed polio teams at relatives’ households within different districts of Karachi and 127 (31.1%) observed polio teams at relatives’ households outside Karachi in different districts of Pakistan and Afghanistan.

Most reported guests came from the east and west districts of Karachi, constituting 25 individuals (28.1%) and 23 individuals (25.8%), respectively. Guests reported from districts outside Karachi included 11 (12.4%) from Quetta, 2 (2.2%) from DI Khan, 1 (1.1%) each from Badin, Peshawar, Hub, Zhob, Qila Saifullah of Pakistan, 4 (4.5%) from Afghanistan, and 1 (1.1%) each from Baghlan and Kunduz (Figure 2, Table 2).

The incidences of travel to Afghanistan without specifying a district name were reported by nine individuals (3.6%). Specifically, six individuals (2.4%) reported travel to Kabul, another six (2.4%) to Kunduz, five (2.0%) to Kandahar, two (0.8%) to Nangarhar, two (0.8%) to Mazar-e-Sharif, and others. Additionally, 28 individuals (11.2%) reported travel to Quetta, 17 (6.8%) to Peshawar, 6 (2.6%) to Zhob, 6 (2.6%) to Hyderabad, 5 (2%) to Pishin, 2 (0.8%) to Waziristan, and 1 (0.4%) each from Badin, Mirpurkhas, Hub, Qila Saifullah, and others (Figure 2, Table 2). Furthermore, travel within different districts of Karachi was high as most traveled to east district (72 (28.8%)), and 26 (10.4%) traveled to west district. Figure 2 provides a visual representation depicting the details of travel to and guest arrivals from various districts within the Karachi division, as well as districts in other regions of Pakistan and Afghanistan.

Fair to moderate agreement was observed between information provided by the household members and FLWs on the variable’s duration of living at current residence (Kappa = 0.370), travel history (Kappa = 0.429), guest arrival (Kappa = 0.395), and household children vaccinated for OPV (Kappa = 0.419), and the agreement was statistically significant for these variables with *p*-value = <0.001. However, no agreement was observed for guest children vaccinated with OPV (Kappa = 0.000, *p*-value = <0.999), but it was also statistically insignificant (Table 2). Furthermore, substantial or perfect agreement was not observed for any variable.

## 4. Discussion

Simultaneous isolation of WPV1 from four different sites of Karachi with genetic linkages related to an ES isolate from Jalalabad, Nangarhar, Afghanistan (AFG) in November 2022, was an alarming situation for the polio program of Sindh, Pakistan [10,18]. All four VP1 genomes differed by <1.5% from the November 2022 isolate. They were isolated within a very short time, and their genomes differed by <0.5%. This is within the margin of error of 1% divergence per year of circulation. The small difference in genomes suggests short-term local circulation; however, there is a possibility that this circulation might still have occurred elsewhere and then been introduced in Karachi along with the extensive population movement to the different catchment areas. Karachi has remained one of the country’s greatest historical poliovirus exporters, with evidence of viruses linked to Karachi isolates appearing in many districts across Pakistan and Afghanistan once Karachi becomes infected [19,20,21]. In the past, genetic sequence data analysis indicated that most WPV1 lineages were transmitted between Pakistan and Afghanistan [22,23,24]. Due to the uncontrolled population movement across the borders between the two countries, the Pakistan–Afghanistan block is considered a single poliovirus reservoir that shares multiple poliovirus lineages as determined from the genetic data analysis of the past few decades [22,24]. From this study, it is evident there is heavy population movement among Afghan household members, with 26% of household members reporting recent travel within Karachi and from WPV1-infected areas of Pakistan and Afghanistan. Also, 56% of participants observed polio teams at relatives’ households within different districts of Karachi and 31% at relatives’ households outside Karachi in different districts of Pakistan and Afghanistan. In addition, 34% of guests arrive from different districts within Karachi, other districts of Pakistan, and Afghanistan. This suggests extensive population movement from Afghanistan to Pakistan. The virus is traveling into Pakistan from Afghanistan, as a majority of the WPV1 found in ES of Punjab, KP, and Karachi originated from Afghanistan [25]. In the current scenario, it is an import in Karachi, due to the high-risk population movement from Afghanistan to Pakistan [25]. In a previous study, a high number of high-risk and mobile populations (HRMP) with substantial links with Afghanistan and throughout Pakistan were also reported overall; 84% of children originated outside of their current district, including 29% from Afghanistan [26].

Extensive population movement within different districts of the Karachi division and across Pakistan in the six months preceding the study was also observed. This includes movement to districts that have already reported polioviruses such as Peshawar and Waziristan. We have also noted reported population movement to Hyderabad, Mirpurkhas, Pishin, Hub, and Quetta districts that should be considered the next destinations of poliovirus spread within the country. Karachi does not share borders with Afghanistan and the distance from Karachi to different travel sites is as far as Peshawar (1554.5 km), Waziristan (1181.1 km), Pishin (735.8 km), Quetta (685.4 km) Mirpurkhas (232.8 km), and Hyderabad (163.7 km) and as close as Hub (22.9 km). Epidemiological analysis of WPV1 shows that in the past, most of the poliovirus burden was shared by three major reservoirs including Karachi, Peshawar, and Quetta block (64.2% in 2015, 75.4% in 2016, and 76.7% in 2017) in Pakistan [21,27]. The past genetic data reflect sustained transmission within reservoir areas, further expanded by periodic importations, which is also evident from population movement patterns within core reservoir areas as identified in this study. Before the isolation of these four WPV1 from Karachi, only two ES samples came back positive for WPV1 from Karachi, but after this event, several ES sites became positive in Karachi and all were linked to each other. A total of 32 ES isolates were reported from Karachi in the year 2023. Moreover, two human-confirmed polio cases were also reported from Karachi in October 2023.

The United Nations estimates that between 3.7 and 4.4 million Afghans reside in Pakistan [28]. Sindh province has 73,789 registered Afghan citizens with most living in Karachi [28]. In October 2023, the interior minister of Pakistan ordered all undocumented Afghans to voluntarily leave the country by 1st November 2023, which impacted the approach to conducting this survey. Due to these repatriation activities initiated by the government of Pakistan, the investigators anticipated that the Afghan population could be reluctant to give information about their origin, travel, and guest arrival from Afghanistan. Therefore, the questionnaire included questions like whether participants observed polio team visits at relatives’ houses in other districts of Pakistan and Afghanistan to give an indirect indication of the travel of the household members to cross-check with directly reported travel information. We also selected the survey timeline carefully to avoid conflict between Afghan household members and polio workers in the same area due to multiple knocks on their doors within a short timeframe. The survey was conducted in the second week of November after the completion of the outbreak response campaign and after the deadline for repatriation so that Afghan families would not speculate the polio worker was connected to the repatriation activities to avoid negatively impacting the polio campaign. More than 80% of families lived in a place for more than 5 years, but they have also maintained their connections with their hometowns in Afghanistan. They are traveling to Afghanistan, and guests from Afghanistan are also coming to visit them. This frequent travel in and out of the country is associated with the introduction of WPV1 in Karachi.

It is encouraging to know from the household members that the behavior of the polio team with the families was good as stated by 97%, and only 2% were not happy with the behavior of FLWs. The satisfactory behavior of the polio team was also reported by the majority of participants in a previous study conducted in high-risk unions in Karachi [14]. Fair to moderate agreement was observed between information provided by the household members and FLWs on the duration of living at the current residence, travel history, guest arrival, and household children vaccinated for OPV. However, no agreement was observed for guest children vaccinated with OPV. Also, substantial or perfect agreement was not observed for any information the household members and FLWs provided. It indicates that Afghan household members are less likely to give complete or accurate information to FLWs when asked about travel or guests.

Contrary to our hypothesis that the polio vaccination status of children under five years of age belonging to the Afghan population in Karachi is sub-optimal, 97% of children received OPV in the last SIA. High vaccination coverage of 98% in HRMP including Afghan refugees was also reported in a previous study [26]. Vaccination of children of HRMP is a priority of the polio program in Pakistan. Household members also indicated that 23% of guest children were also vaccinated by the polio teams at their homes. However, we cannot rule out recall bias or deliberate overreporting of polio vaccination status by the household members. It is also a limitation of our study that we have not asked about the age of the household member who traveled to Afghanistan as a person of more than 5 years old who was vaccinated or naturally exposed to poliovirus might still be an asymptomatic carrier. Reported routine immunization status showed a suboptimal level of vaccination. A total of 57% of respondents reported children in their households had received all the vaccines on the routine immunization schedule, 39% reported partial vaccination, and 4% reported their children had not received any routine vaccines. Earlier studies conducted in Karachi also report a low proportion of fully immunized children [29]. Another limitation is that we have not collected information separately on each vaccine of the RI in the questionnaire, therefore the proportion of fully immunized children may be overreported. The phenomenon of over-reporting vaccination rates has been documented across various contexts, revealing discrepancies between reported and actual coverage levels. For instance, research in India has highlighted discrepancies between reported and verified immunization coverage rates, emphasizing the necessity for robust verification mechanisms [30]. In Pakistan, evaluations of polio vaccination campaigns have also encountered challenges in accurately assessing coverage due to factors such as incomplete reporting and logistical constraints during door-to-door campaigns [31]. These examples underscore the critical importance of implementing independent verification mechanisms to validate reported data, thereby ensuring accurate assessment and facilitating effective public health responses. One of the limitations is the timing of data collection immediately after a vaccination campaign and during a politically sensitive period, such as the repatriation of Afghans, might have influenced participant responses. Despite the reported high coverage from the polio program’s door-to-door OPV vaccination campaigns, the presence of WPV detections in ES sites across Karachi confirms that unvaccinated or under-vaccinated carriers of WPV1 exist in Karachi that are shedding poliovirus in the environment. Several factors may contribute to this discrepancy. Migrants or visitors who are not immunized might introduce or reintroduce the virus to the local population. Furthermore, inaccurate reporting and record-keeping can lead to over-reporting of vaccination coverage due to systemic inaccuracies, social desirability bias, or record-keeping errors, giving a false sense of security regarding the actual immunization status of the population. Also, certain fully vaccinated asymptomatic children can be carriers of poliovirus. Nevertheless, there is a possibility of vaccination failures as reported in previous studies conducted in Uttar Pradesh and Behar, India [32,33]. Even some children with 30 doses still were paralyzed as children were vaccinated with OPV but the vaccination was not as effective as expected or OPV did not reach optimal titers due to malnutrition status, overcrowding, high amounts of poliovirus in the environment, and competition by other enteric viruses [32,33].

To determine the root causes of continued poliovirus transmission despite reported high vaccination coverage, several approaches can be employed. First, seroprevalence studies measure the actual immunity levels in the population by collecting and analyzing blood samples from a representative sample of both children and adults to assess the presence of poliovirus antibodies. Second, molecular epidemiology traces the source and transmission pathways of the virus through genetic sequencing of poliovirus isolates from infected individuals or environments. Identifying genetic similarities can help pinpoint whether it is due to local transmission or importation from other regions. Third, evaluating vaccine efficacy and whether vaccine failure contributes to transmission by comparing immune responses and protection in vaccinated individuals, considering factors such as the number of doses received and the time since vaccination. Identifying potential issues with vaccine efficacy, such as cold chain failures or vaccine handling issues, can help implement corrective measures.

The following policy steps are recommended to enhance poliovirus eradication efforts in Karachi: firstly, targeted vaccination campaigns should prioritize intensive efforts in districts with high Afghan populations and mobility, ensuring comprehensive coverage among children under five years old. Secondly, improving community engagement is essential. This can be achieved through culturally sensitive approaches to communication and trust-building with Afghan communities, addressing vaccine hesitancy, and promoting immunization. Thirdly, integrating vaccination efforts with routine immunization services is imperative. Special attention should be given to improving coverage and equity in immunization services among Afghan children. Lastly, fostering cross-border collaboration with Afghan authorities is essential. This collaboration should aim to synchronize vaccination efforts and enhance monitoring of population movements across the Pakistan–Afghanistan border, thereby boosting overall poliovirus eradication efforts.

## 5. Conclusions

In conclusion, this investigation highlights substantial population mobility observed between Afghanistan and Pakistan and significant movement of the Afghan population within Karachi in the last six months. While the polio campaign demonstrated commendable reported coverage, the suboptimal routine immunization status observed among Afghan children residing in Karachi is concerning. These findings warrant attention and targeted implementation of interventions to enhance and sustain both routine and supplementary immunization efforts within this demographic group. The nuanced understanding of population dynamics and immunization challenges provided by this study is crucial for informing targeted public health policies geared towards mitigating the risks not only of polio but also of infectious diseases of importance in Pakistan such as measles, tetanus, diphtheria, etc., and fostering the holistic well-being of susceptible populations in these regions.

## Figures and Tables

**Figure 1 vaccines-12-01006-f001:**
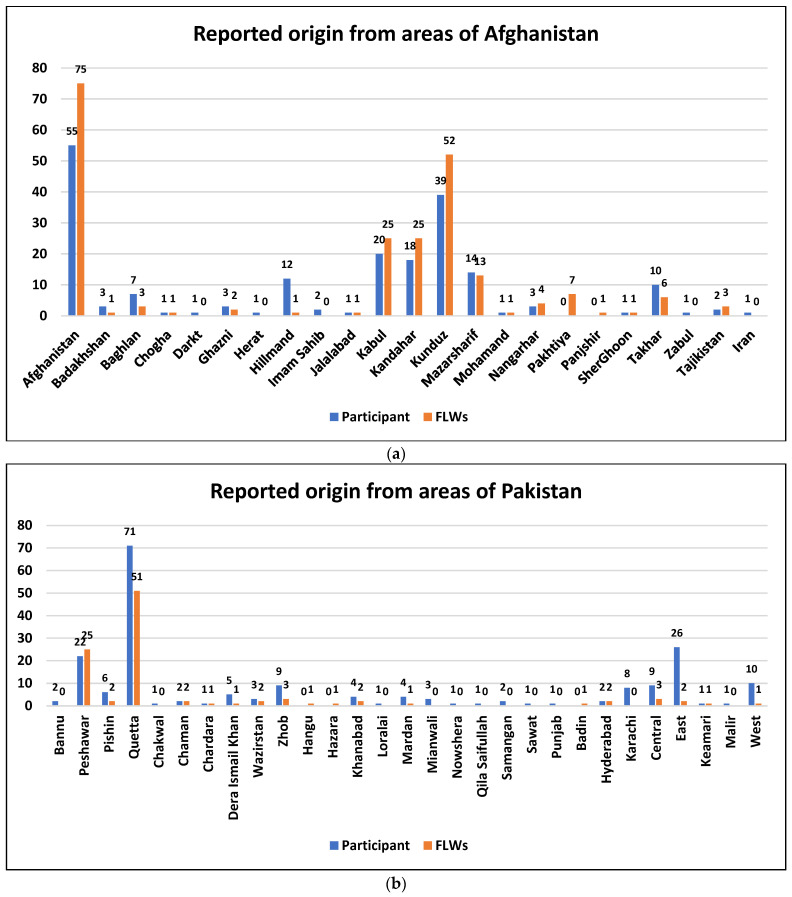
(**a**) Origin of Afghan population (from Afghanistan) before arriving in Karachi, Sindh province, Pakistan. (**b**) Origin of Afghan population (within Pakistan) before arriving in Karachi, Sindh province, Pakistan.

**Figure 2 vaccines-12-01006-f002:**
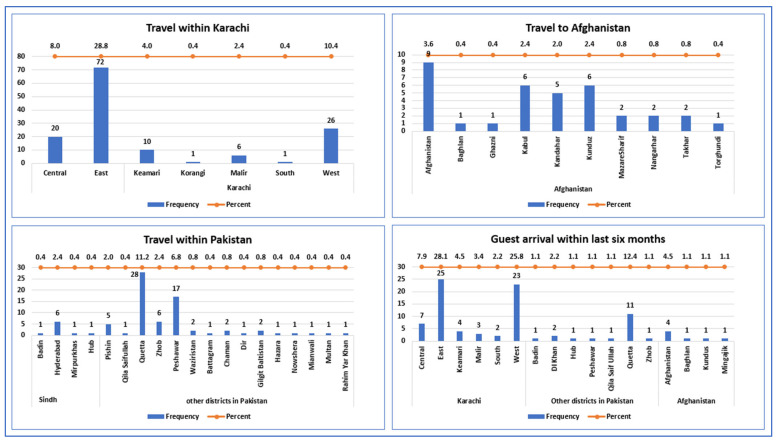
Afghan population movement pattern within Karachi, Sindh province, other districts of Pakistan, and Afghanistan.

**Table 1 vaccines-12-01006-t001:** Characteristics of Afghan study participants in 12 Union Councils of Karachi division, Sindh, Pakistan.

Variable	Number (Percentage) *
Town	
Nazimabad	30 (7.3)
North Nazimabad	21 (5.1)
Gadap	178 (43.5)
Gulshan-e-Iqbal	30 (7.3)
Keamari	30 (7.3)
Orangi	60 (14.7)
SITE	60 (14.7)
Union Councils	
Haji Mureed Goth—44	30 (7.3)
Pahar Ganj—20	21 (5.1)
Gujro A	30 (7.3)
Gujro B	58 (14.2)
Gujro C	29 (7.1)
Safora—13	30 (7.3)
Sultanabad—2	30 (7.3)
Songal—5	61 (14.9)
Baloch Goth—13	30 (7.3)
Mominabad—1	30 (7.3)
Qasba Colony—8	31 (7.6)
Islamia Colony—9	29 (7.1)
Sex of participant	
Female	354 (86.6)
Male	55 (13.4%)
Age (mean ± SD)	36.4363 (11.42686)
Tribe (Top 12)	
Suleman Khel	63 (15.4)
Aka Khel	48 (11.7)
Uzbek	33 (8.1)
Tajik	33 (8.1)
Turkaman	18 (4.4)
Kharoti	13 (3.2)
Kakar	12 (2.9)
Mohmand	9 (2.2)
Noorzai	6 (1.5)
Baloch	5 (1.2)
Farsi	4 (1)
Hazara	3 (0.7)
Language	
Pashto	298 (72.9)
Dari	84 (20.5)
Turkman	20 (4.9)
Uzbek	7 (1.7)
Number of children under five years of age (mean ± SD)	2.342 (1.6375)
Number of children under two years of age (mean ± SD)	0.748 (0.7938)
OPV vaccination status	
No	14 (3.4)
Yes	395 (96.6)
RI status	
No	17 (4.2)
Yes Partial	158 (38.6)
Yes Complete	234 (57.2)
Behavior of polio team FLWs with family	
Good	398 (97.3)
Satisfactory	3 (0.7)
Bad	8 (2)

* Number indicates one response per household and its percentage from total participants.

**Table 2 vaccines-12-01006-t002:** Comparison of information provided by the Afghan study participants and frontline workers in Karachi, Sindh, Pakistan.

Variables	Afghan Household Member Number (%)	Frontline Workers Number (%)	Kappa Statistics	*p*-Value
Duration of living at the current residence			0.370	<0.001
<6 Months	10 (2.4)	19 (4.6)		
6–12 Months	3 (0.7)	8 (2.0)		
1–2 Years	33 (8.1)	42 (10.3)		
3–5 Years	20 (4.9)	24 (5.9)		
5–10 Years	12 (2.9)	34 (8.3)		
10 years or more	331 (80.9)	247 (60.4)		
Don’t know	-	35 (8.6)		
Travel history			0.429	<0.001
No	296 (72.4)	336 (82.2)		
Yes	105 (25.7)	73 (17.8)		
Guest arrival			0.395	<0.001
No	269 (65.8)	334 (81.7)		
Yes	140 (34.2)	75 (18.3)		
Any Guest children vaccinated			0.000	0.999
No	317 (77.5)	329 (80.4)		
Yes	92 (22.5)	80 (19.6)		
OPV vaccination status			0.419	<0.001
No	14 (3.4)	9 (2.2)		
Yes	395 (96.6)	400 (97.8)		
Witnessing polio vaccination at another location in Karachi				
No	179 (43.8)			
Yes	230 (56.2)	-	-	-
Witnessing polio vaccination at another location outside Karachi				
No	282 (68.9)			
Yes	127 (31.1)	-	-	-

## Data Availability

The datasets used and analyzed during the current study are available at the EOC Sindh (sindh.eoc@gmail.com) on a reasonable request.

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
