# Peer review of "Population Movement and Poliovirus Spread across Pakistan and Afghanistan in 2023"

_vaccines, 2024, doi:10.3390/vaccines12091006_

Round 1

Reviewer 1 Report

Comments and Suggestions for Authors

Review: vaccines-3087882-peer-review-v1

“Population Movement and Poliovirus Spread across Pakistan and Afghanistan in 2023”.

Genetically related, wild type 1 polioviruses (WPV1), were isolates from 4 different environmental surveillance (ES) samples with non-overlapping catchment populations in Karachi, Pakistan, within a 7-day interval in September 2023. Such a pattern of WPV1 isolations could be due to local circulation and/or to separate introductions from an external reservoir or reservoirs. This manuscript documents polio vaccine coverage and population movements within members of the Afghan sub-population in Karachi in a cross-sectional study to help rule in or rule out each of the two explanations. Separate introductions from an external reservoir or reservoirs could not be ruled out due to extensive population movement during the 5-month period prior to the Isolation.

General comments:

1.Please indicate the routine polio vaccine immunization (RI) program in Pakistan in the Introduction or Materials and Methods and the supplementary immunization activities (SIAs) including national and sub-national immunization activities. This is needed to better understand the findings on immunization status. Include any information on immunization activities for children entering Pakistan from Afghanistan.

a.        Line 187: “the October Supplementary National Immunization Days (SNIDs) campaign in Karachi.” Was this a SNID or an SIA covering the entire country?

b.        Information on RI and SIAs will help the reader to evaluate Lines 252-255.

c.        Is it possible to relate the ages of the children to the vaccine coverage (for example: A child under the age of 14 weeks would not have received the complete RI schedule, but may have received an additional  dose during an SIA).

d.        In 2022 there were 2 NIDs and 5 SNIDS and in 2023 there were 1 NID, 3 SNIDs and 4 responses to case reports, many of which covered children in Sindh district including Karachi. Were these extra doses of bOPV taken into account when determining whether children were fully vaccinated.

2.There is some confusion throughout the manuscript between use of the terms “persons”, “participants”, and “households”. Specifically, was it one questionnaire/response per household?

3.Lines 73 to 86: This paper does not deal with the molecular analysis of the 4 WPV1 ES isolates.

a.        However, the authors state that all 4 Karachi ES isolates all were genetically related to an ES isolate from Jalalabad, Nangarhar, Afghanistan collected in November 2022: ES-AFG-Nangarhar-Jalalabad-Nov2022. All 4 VP1 genomes differed by <1.5% from the 20922 isolate. This is within the margin of error of 1% divergence per year of circulation. The authors should, if possible, provide a reference that indicates the relatedness of the 4 Karachi isolates to each other. They were isolated within a very short time. A small difference would favor local circulation, however, this circulation might still have occurred elsewhere and then been introduced separately. On the other hand, since the authors document extensive population movement, a significant sequence difference would favor ruling out a single infected person traveling to the different catchment areas. For a short term local circulation, isolates should be very closely related.  For separate introductions, each isolate may be more closely related to isolates from one or more external reservoirs than they are to themselves.

b.        Were there any subsequent genetically related WPV1 isolations in Karachi? A lack of further isolations would indicate that transmission is not extensive. Yet your reference 10 indicates that “This  [Oct Isolate from Karachi East] is the 12th positive sample from Karachi and sixth from Karachi East this [2003] year. The fourth case of the country was also reported from this district earlier this month.

4. Lines 235 to 249. 83.8% of the families lived in-place for >5 years. What is your estimate of the likelihood that members of these families introduced the WPV1 from their previous place of residence?

5.Line 256-257 should be moved to Section 2.3.

6.Lines 264 to 275. Was there any indication of the age of the members who traveled? A person > 5yrs old who was vaccinated or naturally exposed might still be an asymptomatic carrier.

7.Line 266 to 270  and line 396: This “92 (22.5%) participants reported that guest children were vaccinated in their households. For the questions assessing travel of house-hold members, 230 (56.2%) participants observed polio teams at relatives' households within different districts of Karachi and 127 (31.1%) observed polio teams at relatives' households outside Karachi in different districts of Pakistan and Afghanistan” is an important observation and should be reiterated in the Discussion. Studies looking at genetic divergence after SIAs frequently have isolates with very low divergence after a significance post-SIA interval before isolation. Your data is one clear explanation for this. Specifically, the person excreting the isolate was exposed to vaccine elsewhere at a date after the SIA for their place of residence.

8.There were problems with references in the Discussion

a.        Line 312 Ref 9 is incomplete.

b.        Reference 10 indicates more than 4 ES isolates! See #3b above.

c.        Ref 11 and 12 do not support “Karachi has remained the greatest historical poliovirus exporters.

d.        Line 316 Does reference 14 indicate directionality of transmission.

e.        Line 319 Reference 13 goes here.

f.           Line 326 “Punjab, KP and Karachi originated from Afghanistan.” Needs a reference.

9.Line 372. 97% of children received OPV.

a.        Did all complete their RI schedules?

b.        In Discussion relate this to manuscripts on “vaccination failures” in India Uttar Pradesh and Behar. Children were vaccinated with OPV but the vaccination was not as effective as expected due to less than ideal nutrition status, overcrowding, high amounts of polio in the environment, competition by other enteric viruses (OPV doesn’t reach optimal titers). Some children with 30 doses still were paralyzed.

Specific comments:

1. Table 1 Please indicate the meaning of the heading “Number (%)”. See General comment #2.

2.Line 256-257 Method of sample collection is not clear. Information should be in Materials and Methods. CBV and special mobile team strategy are not defined. Ref 8 may be correct but uses different terms.

3.Line 270. Was there an SIA in Afghanistan within the 6 months prior to November for the family to observe a polio team at their relative’s house?

4.Would it be possible to indicate the distances to some of the travel sites from Karachi?

5.Line 351 “indirect indication of travel.” Nice

6.Line 356 “after the deadline” anyone who did not adhere to the requirement would probably be reluctant to report.

7.Line 383 to 385 “Despite the reported high coverage from the polio program’s door-to-door OPV vaccination campaigns, the presence of WPV detections in ES sites across Karachi confirms that unvaccinated or under-vaccinated carriers of WPV1 exist in Karachi that are shedding poliovirus in the environment.” This “unvaccinated or under-vaccinated carriers “ is not necessarily true. Carriers may be fully vaccinated, asymptomatically infected, individuals. Small children are less likely to travel unless the entire family travels, whereas, youth and young adults may travel long distances for work, school, etc.

Author Response

Response to Reviewer 1 Comments

Title: Population Movement and Poliovirus Spread Across Pakistan and Afghanistan in 2023

Summary

Thank you very much for taking the time to review this manuscript.

We are highly grateful to the reviewer for his valuable feedback, we have addressed all the comments, and it has significantly improved the quality of the manuscript.

Please find the detailed responses below and the revisions/corrections highlighted/in track changes in the re-submitted files.

Point-by-point Response to Comments 

General comments:

General comments 1: Please indicate Pakistan's routine polio vaccine immunization (RI) program in the Introduction or Materials and Methods and the supplementary immunization activities (SIAs) including national and sub-national immunization activities. This is needed to better understand the findings on immunization status. Include any information on immunization activities for children entering Pakistan from Afghanistan.

Response comment 1: Thank you for pointing this out. We agree with this comment. Therefore, we have incorporated information on the RI, SIAs, and cross-border vaccination process in the introduction section in line number 68 to 82. RI schedules added the Methods section line numbers 224 to 236 as per the recommendation.

General comment 1a: Line 187 “The October Supplementary National Immunization Days (SNIDs) campaign in Karachi.” Was this an SNID or an SIA covering the entire country?

Response comment 1a: Thank you for pointing this out. It was an outbreak response campaign (OBR) held from October 30 to November 5, 2023, in Karachi, Pakistan. Updated in line number 207-208.

General comments 1b: Information on RI and SIAs will help the reader to evaluate Lines 252-255.

Response comments 1b: The information is added to the Methods section in line number 224-236 as per the recommendation.

General comments 1c: Is it possible to relate the ages of the children to the vaccine coverage (for example: A child under the age of 14 weeks would not have received the complete RI schedule but may have received an additional dose during an SIA).

Response comments 1c: Unfortunately, the information is not collected on each vaccine or antigen in the questionnaire. It is one of the limitations of the study, added in the discussion section line number 450 – 452.

General comments 1d:  In 2022 there were 2 NIDs and 5 SNIDS and in 2023 there were 1 NID, 3 SNIDs and 4 responses to case reports, many of which covered children in Sindh district including Karachi. Were these extra doses of bOPV taken into account when determining whether children were fully vaccinated?

Response comments 1d: In this study routine immunization coverage is classified into fully vaccinated children who received all the recommended doses in EPI. A partially vaccinated child is a child who started to receive the vaccine through a routine immunization program but could not complete the course with recommended vaccines under EPI. An unvaccinated child is A child who did not receive a single dose of vaccines through a routine immunization program. It does not refer to OPV doses received in SIAs.

General comments 2: There is some confusion throughout the manuscript between the use of the terms “persons”, “participants”, and “households”. Specifically, was it one questionnaire/response per household?

Response comments 2: One questionnaire/ response was completed per household in this study. Also, one participant represents one household. The information is incorporated in the Methods section lines 205 – 207 as per the recommendation.

General comments 3: Lines 73 to 86: This paper does not deal with the molecular analysis of the 4 WPV1 ES isolates.

Response comments 3:  Agreed. molecular analysis is not part of this study. It is a limitation of this study. However, we have incorporated information about the source of these findings in line number 101 – 107.

General comments 3a: However, the authors state that all 4 Karachi ES isolates all were genetically related to an ES isolate from Jalalabad, Nangarhar, Afghanistan collected in November 2022: ES-AFG-Nangarhar-Jalalabad-Nov2022. All 4 VP1 genomes differed by <1.5% from the 20922 isolates. This is within the margin of error of 1% divergence per year of circulation. The authors should, if possible, provide a reference that indicates the relatedness of the 4 Karachi isolates to each other. They were isolated within a very short time. A small difference would favor local circulation, however, this circulation might still have occurred elsewhere and then been introduced separately. On the other hand, since the authors document extensive population movement, a significant sequence difference would favor ruling out a single infected person traveling to the different catchment areas. For a short term local circulation, isolates should be very closely related.  For separate introductions, each isolate may be more closely related to isolates from one or more external reservoirs than they are to themselves.

Response comments 3a: All 4 WPV1 genomes differed by <1.5% from the Nov 2022 isolate of Afghanistan and <0.5% different from each other. We have incorporated a few points in the discussion section lines 352 -359. The genetic sequence data analysis along with findings of outbreak investigations are part of another unpublished work of the national polio program team. Therefore, we cannot add more information about the molecular analysis in this study.

General comments 3b: Were there any subsequent genetically related WPV1 isolations in Karachi? A lack of further isolations would indicate that transmission is not extensive. Yet your reference 10 indicates that “This [Oct Isolate from Karachi East] is the 12th positive sample from Karachi and sixth from Karachi East this [2003] year. The fourth case of the country was also reported from this district earlier this month.

Response comments 3b: before this event, only 2 ES samples came positive from Karachi but after this event, several ES sites became positive in Karachi, and all were linked to each other. A total of 32 ES isolates were reported from Karachi in the year 2023. Moreover, 2 human-confirmed polio cases were also reported from Karachi in October 2023.  Added in lines 396-401.

General comments 4: Lines 235 to 249. 83.8% of the families lived in place for >5 years. What is your estimate of the likelihood that members of these families introduced the WPV1 from their previous place of residence?

Response comments 4: Thank you for pointing this out. families lived in a place for >5 years, but they have also maintained their connections with their hometowns in Afghanistan. They are traveling to Afghanistan and guests from Afghanistan are also coming to visit them. This frequent travel in and out of the country is associated with the introduction of WPV1 in Karachi. Added this aspect in the discussion section in lines 418-422.

General comments 5: Lines 256-257 should be moved to Section 2.3.

Response comments 5: Agreed. Line 256-257 moved to Section 2.3 as per the recommendation.

General comments 6: Lines 264 to 275. Was there any indication of the age of the members who traveled? A person > 5yrs old who was vaccinated or naturally exposed might still be an asymptomatic carrier.

Response comments 6: Agreed. A person of age more than 5 years old who was vaccinated or naturally exposed might still be an asymptomatic carrier but we have not asked about the age of household members who traveled to Afghanistan. It is a limitation of our study incorporated in the discussion section in line number 442 - 445.

General comments 7: Line 266 to 270 and line 396: This “92 (22.5%) participants reported that guest children were vaccinated in their households. For the questions assessing travel of household members, 230 (56.2%) participants observed polio teams at relatives' households within different districts of Karachi and 127 (31.1%) observed polio teams at relatives' households outside Karachi in different districts of Pakistan and Afghanistan” is an important observation and should be reiterated in the Discussion. Studies looking at genetic divergence after SIAs frequently have isolates with very low divergence after a significant post-SIA interval before isolation. Your data is one clear explanation for this. Specifically, the person excreting the isolate was exposed to the vaccine elsewhere at a date after the SIA for their place of residence.

Response comments 7: information about observed polio teams inside and outside Karachi is incorporated in the discussion section line numbers 371-374 as per the recommendation.

General comments 8: There were problems with references in the Discussion

Response comments 8: references updated as per the recommendation.

General comments 8a: Line 312 Ref 9 is incomplete.

Response comments 8a: Reference updated as per the recommendation.

General comments 8b: Reference 10 indicates more than 4 ES isolates! See #3b above.

Response comments 8b: Same as response to 3b, before this event, only 2 ES samples came positive from Karachi but after this event, several ES sites became positive in Karachi, and all were linked to each other. Lines 396-401.

General comments 8c: References 11 and 12 do not support “Karachi has remained the greatest historical poliovirus exporters.

Response comments 8c:  TAG report of June 2023 page number 15 Graph 2 shows the Number of Lineages/Chains Exported from One Zone to Another Across Afghanistan and Pakistan, Showing the Greatest Exporters and Importers Based on the Historic Genetic Data of Poliovirus. This image shows the number of transmissions from Karachi and other areas. We have also incorporated another relevant reference in line 363 as per the recommendation.

General comments 8d: Line 316 Does reference 14 indicate the directionality of transmission.

Response comments 8d: Thank you for pointing this out. The information is updated. The lines are rewritten as per the recommendation.

General comments 8e: Line 319 Reference 13 goes here.

Response comments 8e: references added in line 368 as per the recommendation.

General comments 8f: Line 326 “Punjab, KP and Karachi originated from Afghanistan.” Needs a reference.

Response comments 8f: reference incorporated in line 378 as per the recommendation.

General comments 9: Line 372. 97% of children received OPV.

Response comments 9:  information updated in line 435 - 436 as per the recommendation.  

General comments 9a: Did all complete their RI schedules?

Response comments 9a: We have not collected information separately on each vaccine of the RI in the questionnaire, therefore proportion of fully immunized children may be less than the one reported in this study.

General comments 9b: In Discussion relate this to manuscripts on “vaccination failures” in India Uttar Pradesh and Behar. Children were vaccinated with OPV but the vaccination was not as effective as expected due to less than ideal nutrition status, overcrowding, high amounts of polio in the environment, competition by other enteric viruses (OPV doesn’t reach optimal titers). Some children with 30 doses still were paralyzed.

Response comments 9b: Agreed. Incorporated in the discussion section line number 472 – 479 as per the recommendation.

Specific comments:

  1. Table 1 Please indicate the meaning of the heading “Number (%)”. See General comment #2.

Response 1: The number indicates the frequency of one response per household and the proportions of this response from the total participants are given. As described in the notes below the table 1 as per the recommendation.

  1. Line 256-257 Method of sample collection is not clear. Information should be in Materials and Methods. CBV and special mobile team strategy are not defined. Ref 8 may be correct but uses different terms.

Response 2: CBV and SMT strategy are defined in the Methods section line numbers 209-214 as per the recommendation.

  1. Line 270. Was there an SIA in Afghanistan within the 6 months prior to November for the family to observe a polio team at their relative’s house?

Response 3: TAG report June 2023 Graph number 5 shows SIAs implemented in Afghanistan by Modality from Jan 2022 – May 2023. In the year 2023, there were 2 SNIDs in January and May 1 NID in March, and 2 event responses in February and April 2023 (page 17). Graph 15 shows Endorsed Afghanistan and Pakistan SIA Schedules for the Remainder of 2023. In this schedule, 2 SNIDS in July and October and 1 NID in November were planned. (page 38). In the last six months, at least 3 SNIDs were organized from May to Oct 2023 in Afghanistan.

  1. Would it be possible to indicate the distances to some of the travel sites from Karachi?

Response 4: Karachi does not share borders with Afghanistan and distance from Karachi to different travel sites is as far as Peshawar (1554.5 km), Waziristan (1181.1 km), Pishin (735.8 km), Quetta (685.4 km) Mirpurkhas (232.8 km), Hyderabad (163.7 km) and as close as Hub (22.9km). We have incorporated this information in discussion section line number 388 – 391 as per the recommendation.

  1. Line 351: “indirect indication of travel.” Nice

Response 5: We are highly grateful to the reviewer for his valuable feedback.

  1. Line 356 “after the deadline” anyone who did not adhere to the requirement would probably be reluctant to report.

Response 6: Agreed. It is possible that anyone who did not adhere to the requirement would be reluctant to participate or provide accurate information about travel and guests. Therefore, we made all efforts to ensure that household members do not speculate connections of the surveyors and polio teams with other government activities. We selected surveyors who speak their language.  We also added it in the limitation in the discussion section in line numbers 461 -464.

  1. Line 383 to 385 “Despite the reported high coverage from the polio program’s door-to-door OPV vaccination campaigns, the presence of WPV detections in ES sites across Karachi confirms that unvaccinated or under-vaccinated carriers of WPV1 exist in Karachi that are shedding poliovirus in the environment.” This “unvaccinated or under-vaccinated carriers “ is not necessarily true. Carriers may be fully vaccinated, asymptomatically infected, individuals. Small children are less likely to travel unless the entire family travels, whereas, youth and young adults may travel long distances for work, school, etc.

Response 7: Agreed. We have incorporated information about carriers and vaccine failures in the discussion section line number 472 – 479 as per the recommendation.

Reviewer 2 Report

Comments and Suggestions for Authors

This is a well-presented and constructed manuscript. It would be good if the limitations were more clearly outlined.

Also, the authors could mentioned what are the policy steps that should be taken in response to these findings in a prioritised order.

The over-reporting of vaccination rates in this type of studies is not unusual, the authors could have drawn examples from published literature to highlight the scale of divergence between reported and actual that has been observed in the past (e.g., as a worst case scenario in the given population).

Author Response

Response to Reviewer 2 Comments

Title: Population Movement and Poliovirus Spread Across Pakistan and Afghanistan in 2023

Summary

Thank you very much for taking the time to review this manuscript.

We are highly grateful to the reviewer for his valuable feedback, we have addressed all the comments, and it has significantly improved the quality of the manuscript.

Please find the detailed responses below and the revisions/corrections highlighted/in track changes in the re-submitted files.

A point-by-point response to Comments and Suggestions for Authors

Comment 1: It would be good if the limitations were more clearly outlined.

Response 1: Thank you for pointing this out. We agree with this comment. Therefore, we have added limitations of the study in the discussion section line numbers 442 – 445, 450 – 452, 461-463 of the revised manuscript.

Comment 2: The authors could mentioned what are the policy steps that should be taken in response to these findings in a prioritized order.

Response 2: Thank you for pointing this out. We agree with this comment and have added a paragraph on policy steps in the discussion section line numbers 493 – 504 of the revised manuscript.

Comment 3: The over-reporting of vaccination rates in this type of study is not unusual, the authors could have drawn examples from published literature to highlight the scale of divergence between reported and actual that has been observed in the past (e.g., as the worst case scenario in the given population).

Response 3: Thank you for pointing this out. We agree with this comment and have added a paragraph on examples from published literature in the discussion section line numbers 452 – 464, of the revised manuscript.

Reviewer 3 Report

Comments and Suggestions for Authors

The global eradication of poliomyelitis has failed due to the persistence of the viral transmission in Pakistan and Afghanistan, due to inadequate vaccine coverage of children. Wild polio virus type one has been isolated from several sites in Karachi, and match the strains isolated in Afghanistan. The aims of the study were: 1. to identify the origin of the Afghan population and their patterns of movement within Karachi

2. to assess the polio vaccination status of children under the age of five, and 3. to investigate the travel history and guest arrival patterns of individuals from Afghanistan and other regions known to be affected by Wild Poliovirus type 1 (WPV1) within the preceding six months.

The study was carefully designed taking into account the social ethnic characteristics of the population.

It was ethically approved. To assistive reliability of information of them directly from the study subjects, independent information is collected from frontline healthcare workers. 

It was identified that there was substantial population mobility between Afghanistan and Pakistan and significant movement of the Afghan population within Karachi in the previous six months. The reported coverage of children and five polio vaccination was high but there was discordance with high incidence of inadequate coverage by routine child immunisation of the study population.

There was modest, but reassuring agreement between the data directly from the studies subjects and From frontline healthcare workers. It is not surprising that the study subjects had minimal information about the vaccination status of the children belonging to guests. (I cannot think of any population of this type of question from guest guests, who are visiting them irrespective of the circumstances).

Recommendations to authors:

The manuscript would be enhanced if there was a discussion of why there was a discrepancy between the apparent high vaccination status of the children and wild poliovirus transmission in the study area.

It is possible that the transmission was from adults who are nonimmune or alternatively, the information given about the vaccination status of children was unreliable, or alternatively there was vaccine failure. In what way could these possibilities be assessed?

Author Response

Response to Reviewer 3 Comments

Title: Population Movement and Poliovirus Spread Across Pakistan and Afghanistan in 2023

Summary

Thank you very much for taking the time to review this manuscript.

We are highly grateful to the reviewer for his valuable feedback, we have addressed all the comments, and it has significantly improved the quality of the manuscript.

Please find the detailed responses below and the revisions/corrections highlighted/in track changes in the re-submitted files.

Point-by-point Response to Comments 

Comment 1: The manuscript would be enhanced if there was a discussion of why there was a discrepancy between the apparent high vaccination status of the children and wild poliovirus transmission in the study area.

Response 1: Thank you for pointing this out. We agree with this comment. Therefore, we have added a paragraph in the discussion section in line number 467 – 478 of the revised manuscript.

Comment 2: It is possible that the transmission was from adults who are nonimmune or alternatively, the information given about the vaccination status of children was unreliable, or alternatively there was vaccine failure. In what way could these possibilities be assessed?

Response 2: Thank you for pointing this out. We agree with this comment and have added a paragraph the discussion section in line number 480 – 492 of the revised manuscript.